# Resistin Enhances VCAM-1 Expression and Monocyte Adhesion in Human Osteoarthritis Synovial Fibroblasts by Inhibiting MiR-381 Expression through the PKC, p38, and JNK Signaling Pathways

**DOI:** 10.3390/cells9061369

**Published:** 2020-06-01

**Authors:** Wei-Cheng Chen, Chih-Yang Lin, Shu-Jui Kuo, Shan-Chi Liu, Yung-Chang Lu, Yen-Ling Chen, Shih-Wei Wang, Chih-Hsin Tang

**Affiliations:** 1Department of Medicine, MacKay Medical College, New Taipei City 252, Taiwan; wchena7648@gmail.com (W.-C.C.); p123400@hotmail.com (C.-Y.L.); yungchanglu@mac.com (Y.-C.L.); 2Division of Sports Medicine & Surgery, Department of Orthopedic Surgery, MacKay Memorial Hospital, Taipei 104, Taiwan; 3School of Medicine, China Medical University, Taichung 404, Taiwan; b90401073@gmail.com; 4Department of Orthopedic Surgery, China Medical University Hospital, Taichung 404, Taiwan; 5Department of Medical Education and Research, China Medical University Beigang Hospital, Yunlin 651, Taiwan; sdsaw.tw@yahoo.com.tw; 6Department of Fragrance and Cosmetic Science, College of Pharmacy, Kaohsiung Medical University, Kaohsiung 807, Taiwan; yelichen@kmu.edu.tw; 7Graduate Institute of Natural Products, College of Pharmacy, Kaohsiung Medical University, Kaohsiung 807, Taiwan; 8Chinese Medicine Research Center, China Medical University, Taichung 404, Taiwan; 9Department of Biotechnology, College of Health Science, Asia University, Taichung 404, Taiwan

**Keywords:** osteoarthritis, resistin, VCAM-1, miR-381, monocytes

## Abstract

The development of osteoarthritis (OA) is characterized by synovial inflammation and the upregulation of vascular cell adhesion molecule type 1 (VCAM-1) in human osteoarthritis synovial fibroblasts (OASFs). This increase in VCAM-1 expression promotes monocyte adhesion to OASFs. The adipokine resistin is known to promote the release of inflammatory cytokines during OA progression. In this study, we identified significantly higher levels of resistin and CD68 (a monocyte surface marker) expression in human OA tissue compared with in healthy control tissue. We also found that resistin enhances VCAM-1 expression in human OASFs and facilitates the adhesion of monocytes to OASFs. These effects were attenuated by inhibitors of PKCα, p38, and JNK; their respective siRNAs; and by a microRNA-381 (miR-381) mimic. In our anterior cruciate ligament transection (ACLT) rat model of OA, the inhibition of resistin activity prevented ACLT-induced damage to the OA rat cartilage and pathological changes in resistin and monocyte expression. We also found that resistin affects VCAM-1 expression and monocyte adhesion in human OASFs by inhibiting miR-381 synthesis via the PKCα, p38, and JNK signaling pathways. Our clarification of the crucial role played by resistin in the pathogenesis of OA may lead to more effective therapy that reduces OA inflammation.

## 1. Introduction

Osteoarthritis (OA) is a progressive degenerative joint disorder characterized by chronic synovial inflammation, cartilage destruction, joint swelling, and pain [1,2]. The inflammatory process is greatly influenced by the upregulation of vascular cell adhesion molecule type 1 (VCAM-1) in human osteoarthritis synovial fibroblasts (OASFs) [3]. This increase in VCAM-1 expression promotes monocyte adhesion to OASFs, leading to more inflammation in the synovium [3]. The levels of soluble VCAM-1 in human joint synovium are recognized as a strong predictor for the risk of hip and knee joint replacement surgery due to severe OA [4], while higher serum VCAM-1 levels are associated with a greater number of affected joints in hand OA [5]. The synthesis of chondrolytic enzymes and proinflammatory mediators by OASFs promotes synovial inflammation, perpetuating the destruction of cartilage [3]. Synovium-targeted therapy is therefore considered to be an effective means of halting the progression of OA [1].

The adipokine resistin is a 12.5 kDa cysteine-rich polypeptide hormone protein produced by adipocytes in mice, but it is predominantly expressed in macrophages in humans [6]. As the resistin-induced stimulation of human macrophages is known to stimulate proinflammatory cytokines and chemokines, resistin appears to be a worthwhile therapeutic target in chronic OA inflammation [6]. Indeed, significant positive correlations have been observed between OA disease severity and the levels of resistin in the human OA serum, synovial fluid, and tissue samples [7].

Noncoding single-stranded microRNAs (miRNAs/miRs) modify gene expression at the post-transcriptional level. The seed sequence pairs with its complementary sequence mainly within the 3′-untranslated region (3′-UTR) of the target mRNA molecule, suppressing target gene expression [8]. Various miRNAs are involved in the pathogenesis of OA; investigations into the correlation between microRNA expression and OA have determined that miR-9 and miR-24 enhance chondrocyte apoptosis, while miR-21 suppresses chondrocyte autophagy [8]. In rodent models of OA, the modulation of miRNAs in the joint reduces OA disease [9]. Notably, miR-9 promotes interleukin (IL)-6 expression in OA joints by inhibiting the expression of monocyte chemoattractant protein-induced protein 1 (MCPIP-1) in IL-1β-stimulated human chondrocytes, indicating that miRNAs facilitate monocyte adhesion to inflammatory lesions in OA [10].

The rat anterior cruciate ligament transection (ACLT) surgical model of OA results in permanent instability in the articular cartilage knee joint [11]. This model effectively mimics the histopathology and symptoms of OA. In this study, the ACLT model demonstrated that resistin shRNA prevented the ACLT-induced OA pathogenesis of OA and cartilage degeneration, and inhibited monocyte infiltration into the synovium.

Although VCAM-1, resistin, and miRNAs are all involved in monocyte adhesion and OA pathogenesis, it remains unclear as to how they interconnect within adipokine signaling, monocyte adhesion, and OA. Considering the importance of synovium inflammation and monocyte adhesion in OA pathogenesis, our study was designed to delineate the correlation between adipokine signaling, monocyte adhesion, and OA. We speculated that resistin upregulates VCAM-1 expression and stimulates monocyte adhesion to OASFs by mediating miRNA expression.

## 2. Materials and Methods

### 2.1. Materials

Antibodies against resistin (SC-376336), PKCα (H-7) (SC-8393), p-p38 (E-1) (SC-166182), p38 (F-9) (SC-271120), p-JNK (G-7) (SC-6254), and JNK (D-2) (SC-7345) were all bought from Santa Cruz (Dallas, TX, USA). The p-PKC α/βII (T638/641) (9375S) antibody was purchased from Cell Signaling Technology (Danvers, MA, USA). The VCAM-1 (ab134047) antibody was purchased from Abcam (Cambridge, MA, USA). The CD68 (NB100-683) antibody was purchased from Novus Biologicals (Colorado, US, USA), and the β-actin (a5441) antibody was purchased from Sigma (Louis, MO, USA). All ON-TARGETplus siRNAs were purchased from Dharmacon (Lafayette, CO, USA). Cell culture supplements were purchased from Invitrogen (Carlsbad, CA, USA). A Dual-Luciferase^®^ Reporter Assay System was bought from Promega (Madison, WI, USA). qPCR primers and probes, as well as the Taqman^®^ One-Step PCR Master Mix, were procured from Applied Biosystems (Foster City, CA, USA).

### 2.2. Clinical Samples

Serum and synovium samples were obtained from patients with end-stage OA (n = 50) undergoing knee replacement surgery and also from those undergoing arthroscopy after trauma/mechanical patellofemoral syndrome (n = 9) (who served as healthy controls) in China Medical University Hospital, Taichung, Taiwan. The study protocol was approved by the Institutional Review Board (IRB) of China Medical University Hospital (CMUH108-REC3-039), and all methods were performed in accordance with the IRB’s guidelines and regulations. Informed written consent was obtained from all study participants.

Serum from the OA patients (n = 50) and healthy controls (n = 9) was analyzed using a resistin ELISA kit (R&D Systems; Minneapolis, MN, USA), according to the manufacturer’s instructions [12].

### 2.3. Cell Culture

Synovium from the suprapatellar pouch of the knee was obtained from patients undergoing total knee replacement surgery for end-stage degenerative OA. Fresh synovium was minced and digested in a solution of collagenase and DNase. Isolated synovial fibroblasts were filtered through 70 μm nylon filters. Cells were cultured in Dulbecco’s modified eagle medium (DMEM) medium supplemented with 10% fetal bovine serum (FBS), 50 U/mL of penicillin, and 50 μg/mL of streptomycin, as previously described [1,13].

THP-1, a human leukemia cell line of monocyte/macrophage lineage, was obtained from the American Type Culture Collection (Manassas, VA, USA) and cultured in RPMI-1640 medium containing 10% FBS.

### 2.4. Real-Time Quantitative PCR Analysis 

Total RNA was extracted from OASFs by TRIzol; reverse transcription used 1 μg of total RNA transcribed into cDNA by oligo (dT) primers. Real-time quantitative PCR (RT-qPCR) used the Taqman^®^ One-Step RT-PCR Master Mix. All RT-qPCR assays were performed using the StepOnePlus sequence detection system (Applied Biosystems) [14,15].

### 2.5. Western Blot Analysis

Cell lysate was processed by SDS-PAGE electrophoresis then transferred to polyvinylidene difluoride (PVDF) membranes, following the method described in our previous work [16,17]. After blocking the blots with 4% bovine serum albumin, they were treated with primary antibody (1:1000) and then secondary antibody (1:1000). Enhanced chemiluminescent imaging of the blots was performed with the UVP Biospectrum system (UVP, Upland, CA, USA) [18,19,20].

### 2.6. Luciferase Assays

Wild-type and mutant VCAM-1 3′-UTR plasmids were purchased from Invitrogen (Carlsbad, CA, USA). OASFs were co-transfected with 1 μg of plasmid and 0.4 μg of β-galactosidase expression vector. The collected OASFs were dispersed over the 12-well plates and lysed with reporter lysis buffer 24 h after transfection. The Dual-Luciferase^®^ Reporter Assay System was used to determine the extent of luciferase and renilla activities in the cellular extracts, and the relative luciferase activity was quantified by the ratio of luciferase/renilla activity referenced to the ratio of control samples [1,21,22].

### 2.7. Cell Adhesion Assay

THP-1 cells were loaded with BCECF-AM (10 μM) for 1 h at 37 °C in DMEM medium and subsequently washed by centrifugation. OASFs spread on glass coverslips were incubated with resistin then incubated with THP-1 cells at 37 °C for 1 h. Nonadherent THP-1 cells were gently washed away with PBS. The number of adherent THP-1 cells was counted under fluorescent microscopy.

### 2.8. OA Model and Micro-Computed Tomography (Micro-CT) Assessment

The details of the OA model and micro-computed tomography (micro-CT) assessment have been described in our previous publication [23,24]. Male Sprague-Dawley (SD) rats (8 weeks of age, weighing 300–350 g) were chosen for our OA model. The left knee was prepared in a surgically sterile fashion. The ACL fibers were transected with a scalpel, and the entire medial meniscus was excised by medial parapatellar mini-arthrotomy. The joint surface was irrigated with sterile saline solution, and both the capsule and skin were sutured after ACL transection and medial meniscectomy. Ampicillin at 50 mg/kg of body weight was given for 5 days after the surgery. After surgery (Day 0), the rats were divided into 3 groups (n = 8 per group): A control group, an ACLT group, and a resistin shRNA-transfected ACLT group. For 6 weeks, the resistin shRNA-transfected ACLT group was given weekly intra-articular injections of ~7.1 × 10^6^ plaque-forming units (PFU) of resistin shRNA. All the rats were allowed to move freely in plastic cages until necropsy at 10 weeks post-surgery. Rat knee joints were extracted promptly after sacrifice and fixed in 3.7% formaldehyde for micro-CT imaging. All animal procedures were approved and performed in accordance with the guidelines of the Institutional Animal Care and Use Committee of China Medical University (CMUIACUC-2018-015).

### 2.9. Micro Computed Tomography (Micro-CT) Imaging

Isolated rat tibias and femurs were fixed in 4% paraformaldehyde and then 70% ethanol. Samples were scanned using the Bruker SkyScan 2211 nano-CT (Bruker micro-CT, Knotich, Belgium) at a resolution of 8.5 µm. Micro-CT was performed using cameras that scanned over 180 degrees of rotation, with a voltage of 90 kVp, a current of 450 µA (8 watt output), and a 0.5 mm aluminum (Al) filter to prevent beam-hardening artifacts. Image reconstruction was performed using the Instarecon reconstruction software (Bruker-micro-CT, Kontich, Belgium), which was also used to perform the ring artifact and beam-hardening correction.

### 2.10. Micro-CT Analysis Strategy

In brief, reconstructed cross-sections were reorientated, 59 slices (0.5 mm) were selected, and we drew manual regions of interest (ROI) of an irregular anatomical contour in the subchondral trabecular bone region for the medial tibial plateau. Thresholding, ROI selection, and bone morphometric and bone mineral density analyses were all performed using the CTAn software (Version 1.7.1, Bruker microCT, Kontich, Belgium) [23,24].

### 2.11. Immunohistochemistry (IHC) Staining

The knee joints were fixed in 1% formaldehyde, decalcified in 10% EDTA and then dehydrated in ethanol/xylene, following the methods described in our previous work [25]. The sagittal sections (5 μm thick) were cut from the medial side of the knee joint. The sections were stained with primary anti-VCAM-1 and anti-CD68 (1:200) antibodies. Biotin-conjugated goat anti-rabbit immunoglobulin G (IgG) was used as the secondary antibody, and 3, 3’-diaminobenzidine tetrahydrochloride, as the substrate for color development. Some specimens were also stained with Safranin-O/Fast-green or hematoxylin and eosin (H&E).

The Osteoarthritis Research Society International (OARSI) scoring system, adapted for sagittal sections, was used to measure structural cartilage changes in the central weight-bearing area of the medial tibial plateau in all samples [26,27]. In this system, the grade of damage from 0 to 6 is defined as the depth of progression of OA into the cartilage, and the stage of damage is defined as the horizontal extent of cartilage involvement from 0 to 4. The final score is the combined value of grade and stage (score range, 0–24). The grade of synovial membrane inflammation (Grade 0 = no changes, Grade 1 = > 3–4 lining cell layers or slight proliferation of the subsynovial tissue, Grade 2 = > 3–4 lining cell layers and proliferation of the subsynovial tissue, Grade 3 = > 4 lining cell layers and proliferation plus the infiltration of the subsynovial tissue with inflammatory cells, Grade 4 = > 4 lining cell layers and proliferation plus the infiltration of subsynovial tissue with a large number of inflammatory cells). The examination was performed blindly by two observers, and the scores were averaged to minimize observer bias.

### 2.12. Statistics

All statistical analyses were carried out using GraphPad Prism v5.0 (GraphPad Software Inc., San Diego, CA, USA), and all values are expressed as the mean ± S.D. The differences between selected pairs from the experimental groups were analyzed for statistical significance using the paired sample *t*-test for in vitro analyses and one-way ANOVA (two-tail) followed by Bonferroni testing for in vivo analyses. The statistical difference was considered to be significant if the *p*-value was < 0.05.

## 3. Results

### 3.1. Resistin Expression is Increased among OA Patients

The ELISA assay results revealed significantly higher serum concentrations of resistin among OA patients (Figure 1A), and there was substantially higher resistin and CD68 (monocyte surface marker) expression in OA tissue in IHC staining (Figure 1B).

### 3.2. Resistin Increases VCAM-1 Expression and Monocyte Adhesion in Human OASFs

Monocyte adhesion to synovium tissue is an important phenomenon in OA pathogenesis [28]. However, nothing is known about the impact of resistin-enhanced VCAM expression and monocyte adhesion to synovium tissue in OA pathogenesis. In this study, resistin (0, 1, 3, and 10 ng/mL) enhanced monocyte (THP-1 cells) adhesion to OASFs in a concentration-dependent manner, and this phenomenon was effectively attenuated by VCAM-1 antibody (Figure 2A,B, Appendix A). Resistin (0, 1, 3, and 10 ng/mL) enhanced protein synthesis (Figure 2C) and VCAM-1 transcription (Figure 2D) in a concentration-dependent manner. VCAM-1 expression was significantly higher in the OA synovium compared with in normal synovium (Figure 2E). These findings indicate that resistin promotes VCAM-1 expression and monocyte adhesion in human OASFs.

### 3.3. Resistin Promotes VCAM-1 Expression and Monocyte Adhesion via the PKC and PKC-Dependent p38 and JNK Signaling Pathways

PKC signaling plays a key role in cellular functions that are triggered by various stimuli, including resistin [21,29]. To validate the role of PKC in resistin-enhanced VCAM-1 expression and monocyte adhesion, OASFs were pretreated with a PKC inhibitor (GF109203x), a specific PKCα/β inhibitor (Gö6976), and a PKCα siRNA before resistin administration. As shown in Figure 3A–C and Appendix A, the pretreatment of OASFs with GF109203x, Gö6976, or a PKCα siRNA before incubation with resistin significantly inhibited resistin-enhanced monocyte (THP-1) adhesion to OASFs. This phenomenon was effectively attenuated by GF109203x and Gö6976 administration (Figure 3D) or PKCα siRNA transfection (Figure 3E). Resistin stimulated PKCα/β phosphorylation in a concentration-dependent manner, as shown in the Western blot analysis (Figure 3F). These results demonstrate that resistin enhances VCAM-1 expression and monocyte adhesion by stimulating PKC phosphorylation.

It is established that downstream targets for resistin intracellular signaling converge upon the activation of the mitogen-activated protein kinase (MAPK) signaling pathway, which includes p38 and JNK [30]. To investigate whether the p38 and JNK pathways affect resistin-enhanced VCAM-1 expression and monocyte adhesion, we incubated cells with resistin and observed monocyte (THP-1)–OASF adhesion under p38 and JNK pathway blockade. Pretreatment with inhibitors of p38 and JNK or their siRNAs reversed resistin-enhanced monocyte adhesion to OASFs (Figure 4A–C, Appendix A), while resistin-enhanced VCAM-1 transcription was reversed by the inhibitors of p38 and JNK (Figure 4D) and their respective siRNAs (Figure 4E). Western blot analysis revealed that resistin time-dependently stimulated the phosphorylation of p38 and JNK (Figure 4F), and resistin-induced p38 and JNK phosphorylation was reduced by PKC inhibitors (Figure 4G). These findings suggest that resistin facilitates VCAM-1 production and monocyte adhesion to human OASFs via PKC and PKC-dependent p38 and JNK signaling pathways.

### 3.4. Resistin Increases VCAM-1 Expression and Monocyte Adhesion via the Inhibition of MiR-381 Synthesis

Various miRNAs display differential expression patterns between OA and normal knees and are involved in the pathogenesis of OA [13,31]. Using open-source software (TargetScan, miRMap, RNAhybrid, and miRWalk), we determined that miR-381 may interfere with VCAM-1 transcription. Resistin concentration-dependently inhibited miR-381 expression (Figure 5A). To determine whether resistin stimulates VCAM-1 expression and monocyte adhesion by inhibiting miR-381 synthesis, we transfected OASFs with miR-381 mimic and observed reductions in resistin-enhanced VCAM-1 transcription (Figure 5B) and monocyte adhesion (Figure 5C,D, Appendix A). We also constructed a luciferase reporter vector with the wild-type 3′-UTR of the VCAM-1 mRNA (wt-VCAM-1-3’-UTR) and a mutated vector harboring mismatches in the predicted miR-381 binding site (mut-VCAM-1-3’-UTR), to determine whether miR-381 interferes with the transcription of the VCAM-1 gene (Figure 5E). Resistin enhanced the luciferase activity of the wt-VCAM-1-3’-UTR plasmid but not that of the mut-VCAM-1-3’-UTR plasmid (Figure 5K). The miR-381 mimic reversed resistin-enhanced luciferase activity in the wt-VCAM-1-3’-UTR plasmid (Figure 5F), indicating that miR-381 directly suppresses VCAM-1 transcription by binding to the 3’-UTR region of the human VCAM-1 mRNA. In addition, the PKC, JNK, and p38 inhibitors (Figure 5G) and their respective siRNAs (Figure 5H) markedly reversed resistin-suppressed miR-381 transcription. Resistin-enhanced luciferase activity in the wt-VCAM-1-3’-UTR plasmid was reversed by the PKC, JNK, and p38 inhibitors (Figure 5I) and their respective siRNAs (Figure 5J). These results suggest that resistin inhibits miR-381 expression via the PKC, p38, and JNK signaling pathways.

### 3.5. Transfection with Resistin shRNA Ameliorates the Severity of OA

To validate the in vivo role of resistin, we investigated the effects of shRNA-mediated resistin knockdown on OA severity in our ACLT model. ACLT knees were injected with ~7.1 × 10^6^ plaque-forming units (PFU) of resistin shRNA once weekly for 6 weeks, and the differences in weight bearing in the hind paw from 0 to 6 weeks were recorded. The ACLT OA knees demonstrated significantly higher differences of weight bearing in the hind paw from 1 to 6 weeks, and the phenomenon could be reversed by resistin shRNA transfection (Figure 6A). The rats were sacrificed at 10 weeks, and the knee joints were extracted for histological study. Under micro-computed tomography (micro-CT) imaging, resistin shRNA administration restored the integrity of the subchondral bone architecture in the ACLT model (Figure 6B). In comparison to control knees, subchondral bone from the ACLT rats had significantly lower subchondral bone volume (Figure 6C), lower subchondral bone mineral density (Figure 6D), lower bone surface density (Figure 6E), higher trabecular separation (Figure 6F), and lower trabecular numbers (Figure 6G). All of these ACLT-induced effects were reversed by resistin shRNA transfection. Compared with the control samples, those from the ACLT group demonstrated significantly higher OARSI scores and synovial inflammation scores (Appendix A), lower cartilage thickness under Safranin-O staining, and significantly higher expression of resistin, VCAM-1, and CD68 (a monocyte surface marker). ACLT-induced histological changes were reversed by resistin shRNA transfection (Figure 6H; Appendix A).

## 4. Discussion

The pathogenesis of OA is very complex and not well understood, although it is well known that synovium inflammation is integral to the disease process [32,33]. The increased expression of adhesion molecules on the surface of synovial lining cells may attract monocytes to inflammatory sites [34]. In this study, we found that VCAM-1 expression was stimulated by resistin, and this facilitated the adhesion of monocytes to OASFs. We also demonstrated that resistin stimulated VCAM-1 expression and monocyte adhesion to OASFs by inhibiting miR-381 expression via the PKC and PKC-dependent p38 and JNK signaling pathways.

Koskinen et al. have documented high levels of resistin in OA synovial fluid that correlated positively with the proinflammatory interleukin (IL)-6 and with the catabolic factors matrix metalloproteinases MMP-1 and MMP-3 in the synovial fluid, indicating that resistin is detrimental in OA pathogenesis [7]. Similarly, we found significantly higher serum resistin levels among OA patients compared with healthy controls, which were confirmed by IHC data, exhibiting higher levels of resistin expression in OA synovium than in healthy synovial tissue. Our clinical cohort yielded only serum OA samples, so we were unable to measure resistin levels in synovial fluid. Interestingly, a previous paper has described finding elevated levels of serum resistin in patients with primary knee OA compared with in healthy controls [35]. We would be interested in confirming this result in future clinical investigations using OA synovial fluid samples.

During OA pathogenesis, key mediators of the inflammatory process include activated monocytes and macrophages. The increased infiltration of these innate immune cells into the synovium leads to inflammation and synovial damage. Using THP-1 cells as a model for human monocytes in this study, we examined monocyte adhesion to OASFs and the adhesiveness between OASFs and THP-1 cells. Our study clearly demonstrates that VCAM-1 facilitates monocyte adhesion to OASFs and serves as a target protein for resistin. Importantly, although THP-1 monocytes can be differentiated into macrophages, the spectrum of response to activating stimuli of THP-1 macrophages differs from that of blood-derived macrophages, with significant phenotypic variations, including, for instance, decreased phagocytic activity compared with human macrophages [36]. Moreover, macrophages exhibit two primary phenotypes: the classically activated (M1) form, a proinflammatory phenotype, or the alternatively activated (M2) form, which encompasses wound healing and regulatory subsets involving anti-inflammatory properties that are driven by IL-1 plus IL-13, IL-10, or immune complexes [37,38,39]. Importantly, the murine macrophage cell line, RAW 264.7, represents an M1 phenotype so cannot be expected to provide results that are consistent with those from THP-1 cells. Future OA research should use human macrophages to determine whether these produce the same effects as murine macrophages or resistin on the OA joint, similar to the effects of monocyte infiltration.

This study used the rat ACLT model of OA. No OA animal models have, as yet, emerged that fully mimic the pathogenesis of human OA disease. The ACLT model is associated with highly reproducible results and rapidly progressive OA disease [11]. Moreover, the ACLT model is associated with molecular changes in the cartilage, synovial inflammation, and subchondral bone sclerosis, which resemble the characteristics of human OA [11]. All of these parameters mean that this surgical OA model is appropriate for this short-term study. As reflected in Figure 6, the transfection of ACLT knees with shRNA prevented ACLT-induced OA pathogenesis and cartilage degeneration, and inhibited monocyte infiltration into the synovium. The next step in our research plan is to develop a monocyte antibody or small inhibitor that could be used in further experiments to confirm our results observed using resistin shRNA.

PKC acts as an intersector for crosstalk between resistin and VCAM-1. The activation of PKC signaling is an integral part of resistin-mediated inflammation in human macrophages [40]. PKC is also an essential component for advanced glycation end product (AGE)-induced VCAM-1 expression in monocytes. In this study, we found that resistin facilitates PKC phosphorylation, while PKC inhibitors and siRNAs abolished resistin-enhanced VCAM-1 production. PKC inhibitors and siRNAs also effectively mitigated resistin-enhanced VCAM-1 dependent monocyte adhesion to OASFs. The MAPK-family kinases, including p38 and JNK, are critical for controlling cell adhesion and motility [41,42]. Our study demonstrates that p38 and JNK inhibitors and their siRNAs reversed resistin-enhanced VCAM-1 expression and monocyte adhesion, indicating that p38 and JNK are required for resistin-enhanced VCAM-1-dependent monocyte adhesion. We also demonstrate that resistin facilitates p38 and JNK phosphorylation. PKC inhibitors mitigated resistin-facilitated p38 and JNK phosphorylation, suggesting that PKC-dependent p38 and JNK activation is essential for resistin-enhanced VCAM-1 expression and monocyte adhesion.

miRNAs serve as post-transcriptional modifiers of gene expression and are involved in various human diseases, including OA [43]. Pharmacotherapy capable of modifying miRNA expression holds promise for mitigating the OA inflammatory process [43,44]. We utilized open-source miRNA software to predict that miR-381 disrupts VCAM-1 transcription. We also found that resistin inhibits miR-381 synthesis and that the transfection of OASFs with miR-381 mimic mitigates resistin-enhanced VCAM-1 expression and monocyte adhesion. PKC, p38, and JNK inhibitors reversed the resistin-induced inhibition of miR-381 expression, indicating that resistin facilitates VCAM-1 production and monocyte adhesion by reducing miR-381 expression via the PKC, p38, and JNK signaling cascades.

In summary, our study illustrates that resistin increases VCAM-1 expression and promotes monocyte adhesion to OASFs by inhibiting miR-381 synthesis via the PKC, p38, and JNK signaling pathways (Figure 7). These findings could improve our knowledge about the role of OASFs in OA pathogenesis and may influence the design of novel, more effective therapy for OA.

## Figures and Tables

**Figure 1 cells-09-01369-f001:**
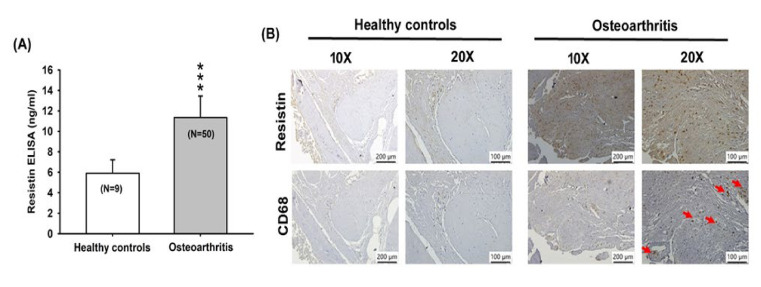
Resistin expression is increased among osteoarthritis (OA) patients. (**A**) Resistin serum levels in healthy controls (n = 9) and OA patients (n = 50) were quantified by the ELISA assay. (**B**) Immunohistochemistry (IHC) staining of resistin and CD68 (monocyte surface marker) for healthy controls (n = 10) and OA synovium (n = 10). *** *p* < 0.001 compared with healthy controls.

**Figure 2 cells-09-01369-f002:**
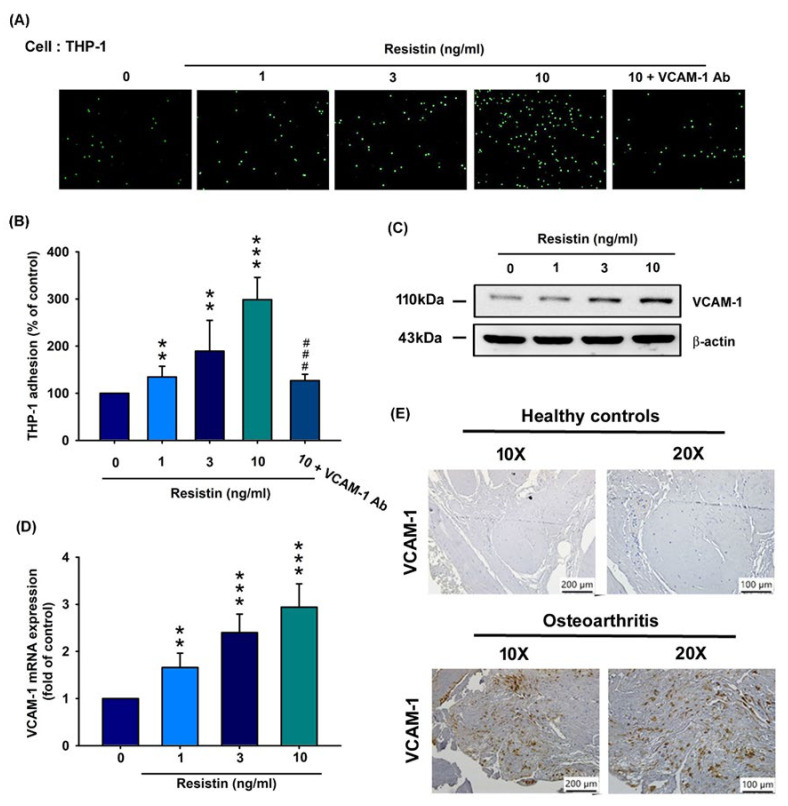
Resistin stimulates VCAM-1 expression and monocyte adhesion to osteoarthritis synovial fibroblasts (OASFs). (**A**–**D**) OASFs were incubated with resistin (0, 1, 3, and 10 ng/mL) only or resistin at 10 ng/mL + VCAM-1 antibody for 24 h. THP-1 cells were subsequently added to OASFs for 1 h. The adherence of THP-1 cells to cultured OASFs was photographed under fluorescence microscopy (n = 4) (**A**) and quantified (**B**). VCAM-1 expression according to varying concentrations of resistin (0, 1, 3, or 10 ng/mL) was quantified by Western blotting (n = 3) (**C**) and RT-qPCR analysis (n = 4) (**D**). (**E**) IHC staining of VCAM-1 for normal (n = 10) and OA synovium (n = 10). ** *p* < 0.01; *** *p* < 0.001 compared with the control group; ### *p* < 0.001 compared with the resistin-treated group.

**Figure 3 cells-09-01369-f003:**
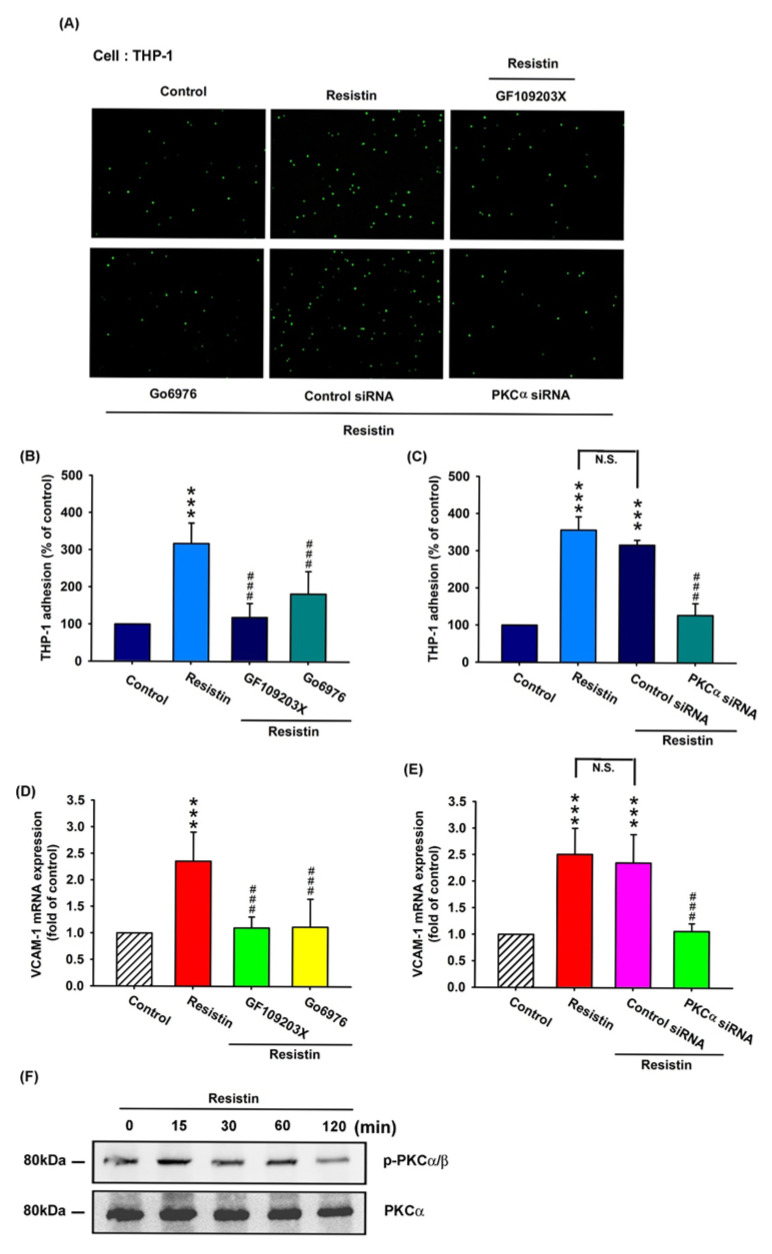
The PKC pathway is involved in resistin-enhanced VCAM-1 expression and monocyte adhesion. (**A**–**E**) OASFs were pretreated with a PKC inhibitor (GF109203x), a specific PKCα/β inhibitor (Gö6976), or a PKCα siRNA, then incubated with varying concentrations of resistin for 24 h. The adherence of THP-1 cells to cultured OASFs was photographed by fluorescence microscopy (**A**) and quantified (n = 4) (**B**,**C**). The transcription levels of VCAM-1 were quantified by the RT-qPCR assay (n = 4) (**D**,**E**). The extent of phosphorylation of PKCα/β under resistin (10 ng/mL) stimulation (for 0, 15, 30, 60, or 120 min) was quantified by Western blotting (n = 3) (**F**). *** *p* < 0.001 compared with control group; ### *p* < 0.001 compared with the resistin-treated group.

**Figure 4 cells-09-01369-f004:**
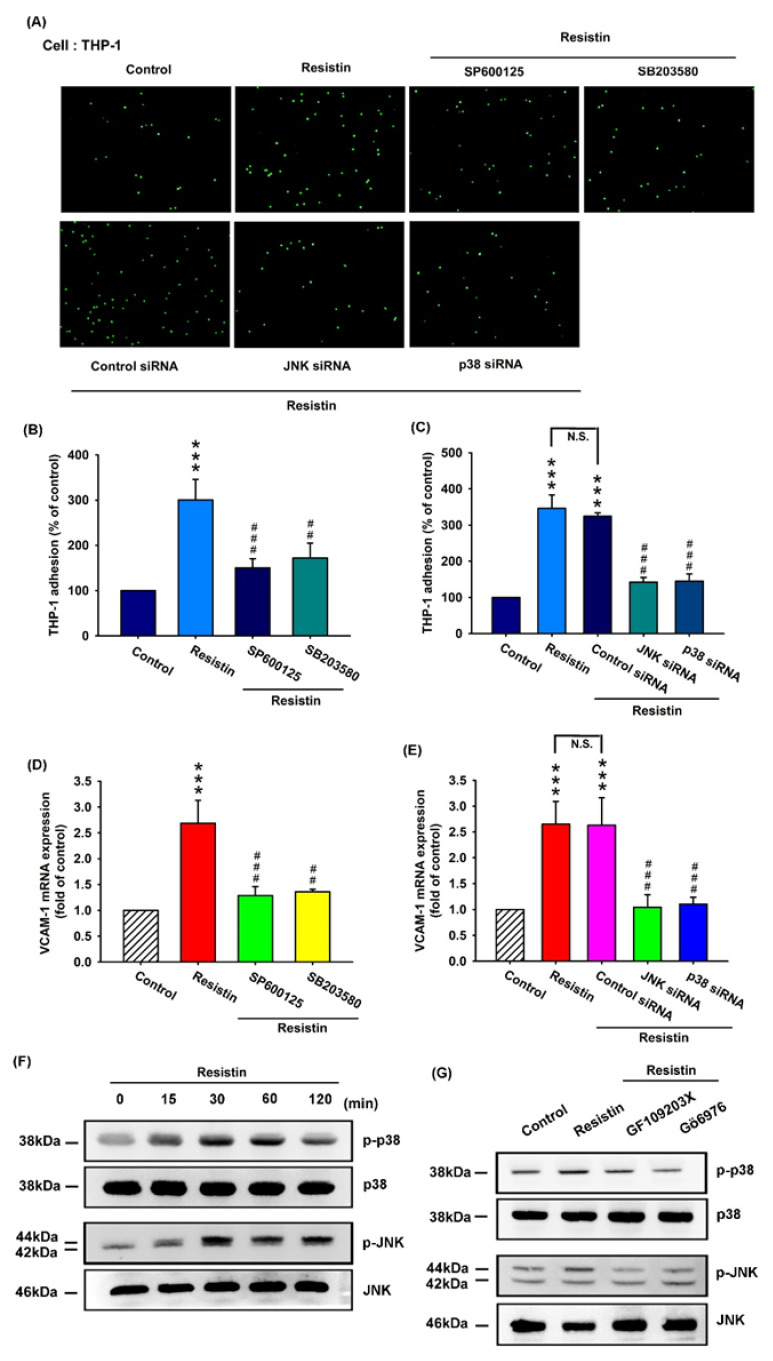
The p38 and JNK pathways are involved in resistin-enhanced VCAM-1 synthesis and monocyte adhesion. (**A**–**E**) OASFs were pretreated with inhibitors of JNK (SP600125) and p38 (SB203580) and their respective siRNAs, then incubated with resistin for 24 h. The adherence of THP-1 cells to cultured OASFs was photographed by fluorescence microscopy (**A**) and quantified (n = 4) (**B**,**C**). The transcription levels of VCAM-1 were quantified by the RT-qPCR assay (n = 4) (**D**,**E**). The extent of phosphorylation of p38 and JNK under resistin (10 ng/mL) stimulation (for 0, 15, 30, 60, or 120 min) was quantified by Western blotting (n = 3) (**F**). (**G**) OASFs were pretreated with a PKC inhibitor (GF109203x) or a specific PKCα/β inhibitor (Gö6976), then incubated with resistin for 24 h. The extent of p38 and JNK phosphorylation was quantified by Western blotting (n = 3). *** *p* < 0.001 compared with the control group; ## *p* < 0.01, and ### *p* < 0.001 compared with the resistin-treated group.

**Figure 5 cells-09-01369-f005:**
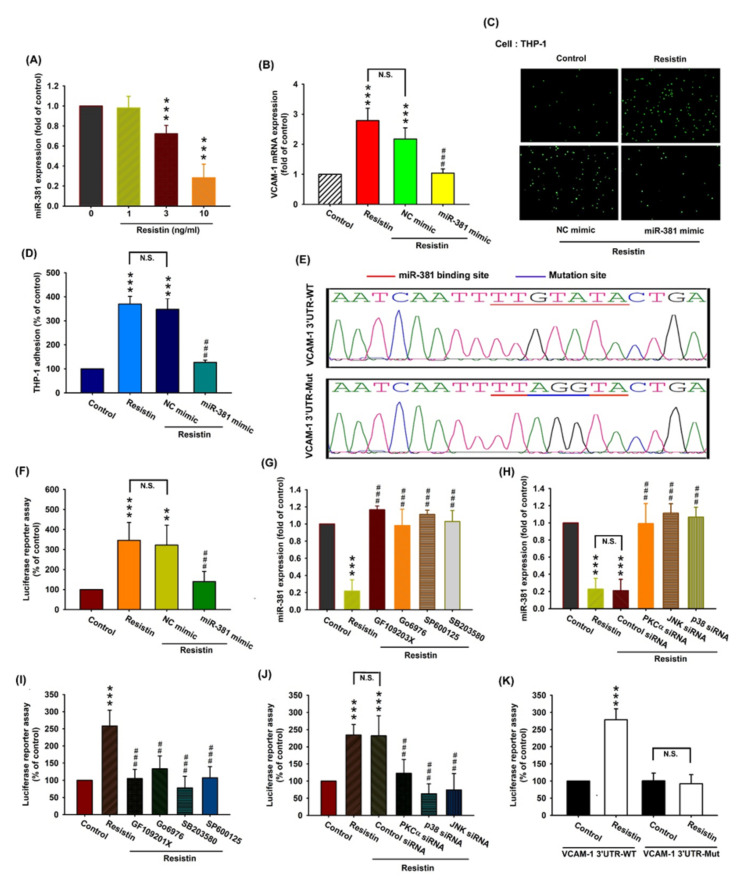
Resistin promotes VCAM-1 expression and monocyte adhesion by suppressing miR-381 expression. (**A**) OASFs were incubated with resistin (0, 1, 3, or 10 ng/mL), and the extent of miR-381 expression was quantified by the RT-qPCR assay. (**B**–**D**) OASFs were transfected with miR-381 mimic or control siRNA, then stimulated with resistin. The extent of VCAM-1 transcription was examined by the RT-qPCR assay (n = 4) (**B**). THP-1 cells were subsequently added to OASFs for 1 h. The adherence of THP-1 cells to cultured OASFs was photographed by fluorescence microscopy and quantified (n = 4) (**C**, **D**). (**E**) Schematic demonstration of wt-VCAM-1 3′-UTR and mut-VCAM-1 3′-UTR. MiR-381 bound with wt-VCAM-1 3′-UTR but not mut-VCAM-1 3′-UTR. (**F**) OASFs were transfected with the wt-VCAM-1-3’-UTR plasmid with or without miR-381 mimic, then stimulated with resistin. Relative luciferase activity was quantified (n = 5). (**G**, **H**) OASFs were pretreated with inhibitors of PKC, JNK, and p38 (n = 4) (**G**) and their respective siRNAs (n = 5) (**H**), then incubated with resistin for 24 h. miR-381 expression levels were quantified by the RT-qPCR assay. (**I**, **J**) OASFs were treated with inhibitors of PKC, JNK, and p38 (n = 5) (**I**) or their respective siRNAs (n = 5) (**J**) and transfected with the wt-VCAM-1-3’-UTR plasmid before being incubated with resistin for 24 h. Relative luciferase activity was quantified. (**K**) OASFs were transfected with the wt-VCAM-1-3’-UTR plasmid or mut-VCAM-1-3’-UTR plasmid, then stimulated with resistin for 24 h. Relative luciferase activity was quantified (n = 5). ** *p* < 0.01, and *** *p* < 0.001 compared with control group; ## *p* < 0.01, and ### *p* < 0.001 compared with the resistin-treated group.

**Figure 6 cells-09-01369-f006:**
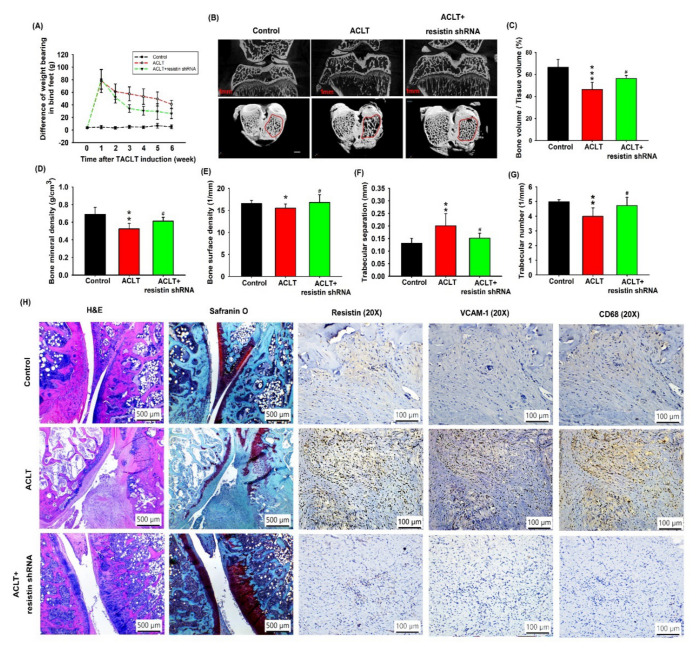
Resistin shRNA ameliorates the histologic severity of OA. (**A**) Between-group differences for changes in weight bearing in each hind paw for the control knee, anterior cruciate transection (ACLT) knee, and resistin shRNA-transfected ACLT knee (n = 8). (**B**) Micro-CT images of the control knee, ACLT knee, and resistin shRNA-transfected ACLT knee (n = 3). (**C**–**G**) The micro-CT parameters, including the subchondral bone volume (**C**), subchondral bone mineral density (**D**), bone surface density (**E**), trabecular separation (**F**), and trabecular number (**G**) of the control knee, ACLT knee, and resistin shRNA-transfected ACLT knee (n = 6). (**H**) Specimens from the control knee, ACLT knee, and resistin-shRNA-transfected ACLT knee were immunostained with Safranin-O, resistin, VCAM-1, and CD68 (monocyte surface marker) (n = 6). * *p* < 0.05, and ** *p* < 0.01 compared with control knees; # *p* < 0.05 compared with the ACLT group.

**Figure 7 cells-09-01369-f007:**
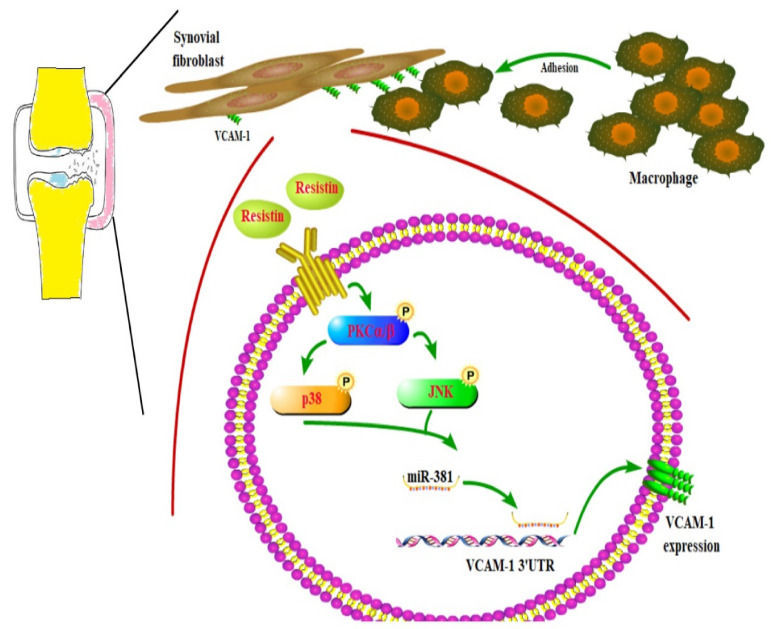
Schematic diagram summarizing the mechanism whereby resistin upregulates VCAM-1 expression in human synovial fibroblasts. Resistin induces VCAM-1 expression and monocyte adhesion through the PKC, p38, and JNK signaling cascades, and reduces miR-381 expression in OASFs.

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
