# Peer review of "Resistin Enhances VCAM-1 Expression and Monocyte Adhesion in Human Osteoarthritis Synovial Fibroblasts by Inhibiting MiR-381 Expression through the PKC, p38, and JNK Signaling Pathways"

_cells, 2020, doi:10.3390/cells9061369_

Round 1

Reviewer 1 Report

This study investigated the adhesion of human monocytes (THP-1 cells) to human synovial fibroblasts (OASFs) obtained from osteoarthritis (OA) patients under the influence of the adipokine resistin. The authors demonstrated that resistin enhances the expression of VCAM-1 in human OASFs, which in turn led to enhanced adhesion of monocytes. Using signaling pathway inhibitors as well as respective siRNAs, the authors identified that PKC, p38, and JNK mediate the resistin mediated effects on monocyte adhesion. Furthermore, the treatment of OASF with miR-381 resulted in attenuation of resistin-dependent increase of VCAM-1 expression and monocyte adhesion. In addition, the authors performed experimental OA in rats using the anterior cruciate transection method and transfected one group with resistin shRNA. The authors state that this treatment is protective, since OA progression was nearly completely inhibited.

In general, the topic is very interesting and up-to-date. However, there are many limitations and points that need improvement or clarification.

The animal part drags the quality of this manuscript down.

  1. Abstract: The first three introductory sentences could be linked somehow. At the moment it is just like a listing of some facts. Sounds a bit awkward. Authors should emphasize the in vivo results, since they have a much higher relevance fort he human disease.
  2. Introduction: the second sentence is strange. Line 47 -suddently, monocyte appear, without describing anything about inflammation in OA and the cells that are involved in synovial inflammation. Line 54-55 – „the clustering of monocytes ist the principial morphological feature“ sounds strange. Please clarify whta you mean by that. Lines 70-75 should be described more simply. Maybe split overlong sentences.Methods
  3.  
  4. Catalogue / ordering numbers and dilutions of antibodies should be provided – the metods shoulb be reproducible for other researchers.
  5. How did the authors isoltae OASF?
  6. What ist he relevance of monocytes in OA? THP-1 cells are non-polarized and not activated – if we think about macrophage subtypes M1 and M2. This point should be at least discussed.
  7. 2.3. Cilinical samples should come first and then the cell culture.
  8. 2.3. What does „non-OA causes“ mean? Knee-trauma patients? Please clarify.
  9. 2.8. Why did the authors only analyze subchondral bone regarding OA symptoms? What about cartilage degeneration scoring (e.g. OARSI score) and synovitis (scoring is missing)?
  10. Few words about animal model should be provided in the introduction. This is completely missing. Benefits and disadvantages?
  11. 2.9. The t-test alone is not sufficient. There are multiple comparisons in almost all figures. Therefore, ANOVA or ANOVA on ranks should also be performed.

Results

  1. 3.1. Lines 161-162 belong tot he introduction and are not necessary here.
  2. 3.1. Did the authors by any chance analyze synovial fluid resistin concentations? Did a previous study analyze resistin in the serum and synovial fluid from the same patient? This should be at least discussed.
  3. Fig. 1A/B What does „normal“ mean? It suggests that the authors analyzed samples from healthy donors, which is not true. Please clarify/define this term.
  4. Fig. 1B The indicated squares in the left panels do not represent the same are as shown in the right panels. The same for Fig.2.
  5. Please indicate the number of samples (n=) and exact p values in all figure legends.
  6. Fig. 2A the THP-1 panels are very dark and the cells are difficult to see. Also the merged images are not ideal. It is not possible to see the OASFs and to distinguish between OASF and THP-1 cells. The same is true for Fig. 3, Fig. 4 and Fig. 5.
  7. Fig. 6 A-G Axis labeling is too small, it is not possible to read it. Similarly, the pics in H are small and the specific positive staining are almost invisible.
  8. Fig. 6B Authors should indicate the region/volume of interest in the subchondral bone.
  9. Fig. 6H What do the H&E and Safranin-O pictures show? Are that sagittal or frontal sections? Do we see madial or lateral knee side? Which area oft he synovium was selected (adjacent to meniscus or to bone)?
  10. Figure 6 legend the second F should be H, i guess (line 292).Discussion
  11.  
  12. Very short. Almost no or only very few comparisons with other studies. Animal model is not discussed at all.

Author Response

Dear Editor

We greatly appreciate the comments of your Reviewers on our manuscript Resistin enhances VCAM-1 expression and monocyte adhesion in human osteoarthritis synovial fibroblasts by inhibiting miR-381 expression through the PKC, p38 and JNK signaling pathways (Ref. No. cells-804877). We have carefully revised the manuscript according to the suggestions raised by the Reviewers, and specific points have been addressed as below. We have used red font to mark up the changes in our Word file.

Reviewer 1

Q1. Abstract: The first three introductory sentences could be linked somehow. At the moment it is just like a listing of some facts. Sounds a bit awkward. Authors should emphasize the in vivo results, since they have a much higher relevance for the human disease.

A: We have rephrased the Abstract.

Q2. Introduction: the second sentence is strange. Line 47 -suddently, monocyte appear, without describing anything about inflammation in OA and the cells that are involved in synovial inflammation. Line 54-55 – “the clustering of monocytes is the principal morphological feature” sounds strange. Please clarify what you mean by that. Lines 70-75 should be described more simply. Maybe split overlong sentences.

A: We have rephrased the first paragraph of the Introduction.

Methods

Q3. Catalogue / ordering numbers and dilutions of antibodies should be provided – the methods should be reproducible for other researchers.

A: We have accordingly provided detailed information about the antibodies in the 2.1. Materials and 2.5. Western blot analysis section.

Q4. How did the authors isolate OASFs?

A: We have provided details about the isolation process in the Materials and Methods section. (lines 130-135)

Q5. What is the relevance of monocytes in OA? THP-1 cells are non-polarized and not activated – if we think about macrophage subtypes M1 and M2. This point should be at least discussed.

A: This aspect has been discussed in the Discussion section. (lines 425-442)

Q6. 2.3. Clinical samples should come first and then the cell culture.

A: We have accordingly moved the clinical samples section to 2.2, which is now followed by section 2.3 describing the cell culture.

Q7. 2.3. What does “non-OA causes” mean? Knee-trauma patients? Please clarify.

A: We have amended non-OA to read as healthy controls. Accordingly, we have also changed the description of the patients without OA to read as: “also from those undergoing arthroscopy after trauma/mechanical patellofemoral syndrome (who served as healthy controls) in China Medical University Hospital”. (lines 118-122)

Q8. 2.8. Why did the authors only analyze subchondral bone regarding OA symptoms? What about cartilage degeneration scoring (e.g. OARSI score) and synovitis (scoring is missing)?

A: We have added information about these aspects accordingly. (lines 212-224) (Supplementary data Fig. S2A&B)

Q9. Few words about animal model should be provided in the introduction. This is completely missing. Benefits and disadvantages?

A: We have added information about the animal model accordingly. (lines 87-91)

Q10. 2.9. The t-test alone is not sufficient. There are multiple comparisons in almost all figures. Therefore, ANOVA or ANOVA on ranks should also be performed.

A: We have accordingly provided more information about the statistical analyses that were performed in Method section (2.12. Statistics).

Results

Q11. 3.1. Lines 161-162 belong to the introduction and are not necessary here.

A: This text has been deleted.

Q12. 3.1. Did the authors by any chance analyze synovial fluid resistin concentrations? Did a previous study analyze resistin in the serum and synovial fluid from the same patient? This should be at least discussed.

A: This aspect has been addressed in the Discussion section. (lines 414-424)

Q13. Fig. 1A/B What does “normal” mean? It suggests that the authors analyzed samples from healthy donors, which is not true. Please clarify/define this term.

A: “Normal” has been changed to “healthy controls”.

Q14. Fig. 1B The indicated squares in the left panels do not represent the same are as shown in the right panels. The same for Fig.2.

A: The images have been corrected (Fig. 1B and 2E).

Q15. Please indicate the number of samples (n=) and exact p values in all figure legends.

A: The n numbers and p-values have been added to all figure legends.

Q16. Fig. 2A the THP-1 panels are very dark and the cells are difficult to see. Also the merged images are not ideal. It is not possible to see the OASFs and to distinguish between OASF and THP-1 cells. The same is true for Fig. 3, Fig. 4 and Fig. 5.

A: (i) Higher-quality images have been provided. (ii) The merged images have been moved to Supplementary Fig. S1.

Q17. Fig. 6 A-G Axis labeling is too small, it is not possible to read it. Similarly, the pics in H are small and the specific positive staining are almost invisible.

A: (i) The labeling has been corrected. (ii) We have accordingly provided higher-quality, larger-sized images in Supplementary Fig. S2.

Q18. Fig. 6B Authors should indicate the region/volume of interest in the subchondral bone.

A: We have now added details about this aspect in 2.10 Micro-CT analysis strategy section (lines 198-202).

Q19. Fig. 6H What do the H&E and Safranin-O pictures show? Are that sagittal or frontal sections? Do we see medial or lateral knee side? Which area of the synovium was selected (adjacent to meniscus or to bone)?

A: (i) Hematoxylin and eosin (H&E) staining is the most widely used staining method in histology and allows localization of nuclei and extracellular proteins. Safranin-O staining is a basic dye that stains growth plate cartilage and articular cartilage.

(ii & iii) Sagittal sections (5 µm thick) were cut from the medial side of the knee joint.

(iv) We selected the synovium region adjacent to the bone.

Q20. Figure 6 legend the second F should be H, I guess (line 292).

A: Figure 6 (F) has been corrected to (H). (line 399)

Q21. Discussion, Very short. Almost no or only very few comparisons with other studies. Animal model is not discussed at all.

A: The Discussion section has been rewritten and lengthened appropriately.

We sincerely hope that our revised manuscript is now suitable for publication in the Cells journal.

Kind regards,

Chih-Hsin Tang, PhD.

Reviewer 2 Report

The article is original and the topic is within the scope of the Cells. However, Authors didn't draft the paper well and I cannot recommend the publication of the article in its present form in such high-ranked journal as Cells. The Authors should rewrite the manuscript before the re-submission of the work.

The main issue with the submitted manuscript is the quite poorly written materials and methods section, which must be substantially improved.

Just to name the few:

The information about IHC staining or ELISA assays is completely missing in this section – the procedures are not described, there is no information that serum was subjected to elisa assay of resitin and synoval tissue samples were subjected to IHC for resisitin, CD68 and VCAM-1. There is no information about what was marked with Western blot. There is no information about the number of repetitions of each determinant. The information about the number of OA and control patients (sec. 2.3) is also missing. The information about the approval of procedures involving rats by ethics committee is missing (approval number). Were control rats injected with some kid of placebo? For uCT, there is no information about ROIs (their location, area) or determined microstructural traits. During statistical analyses, was normal distribution of the data verified? Was two-tailed or one-tailed t-test applied?

I know that the Authors can better (10.3390/cells9040927).

Minor comments (not all of them)

L72 correct to "anterior cruciate ligament transectiot"

L79 Santa Cruz is a city in CA. Santa Cruz Biotechnology headquarters are located in Dallas, TX.

L88 DMEM While it is well-known basal medium all other abbreviations are explained.

L94 synovial tissue, synovium - please unify the nomenclature in whole manuscript

L156 remove extra “mean”

L168 and others: give the exact values of concentration of resistin (1, 3, 10)

Fig. 1A – change the colour of bar of the OA group.

Fig 3F, 4F - please mark on the graphs that these values (0-120) show time of stimulation

Fig 6D unit for BMD is missing

L306 Koskien et al ? ref [7] ?

L351 references are not formatted according to journal requirements

L359 complete the reference data (article no 1505)

Author Response

Dear Editor

We greatly appreciate the comments of your Reviewers on our manuscript Resistin enhances VCAM-1 expression and monocyte adhesion in human osteoarthritis synovial fibroblasts by inhibiting miR-381 expression through the PKC, p38 and JNK signaling pathways (Ref. No. cells-804877). We have carefully revised the manuscript according to the suggestions raised by the Reviewers, and specific points have been addressed as below. We have used red font to mark up the changes in our Word file.

Reviewer 2

Q1. The information about IHC staining or ELISA assays is completely missing in this section – the procedures are not described, there is no information that serum was subjected to Elisa assay of resistin and synovial tissue samples were subjected to IHC for resistin, CD68 and VCAM-1. There is no information about what was marked with Western blot. There is no information about the number of repetitions of each determinant. The information about the number of OA and control patients (sec. 2.3) is also missing. The information about the approval of procedures involving rats by ethics committee is missing (approval number). Were control rats injected with some kind of placebo? For micro-CT, there is no information about ROIs (their location, area) or determined microstructural traits. During statistical analyses, was normal distribution of the data verified? Was two-tailed or one-tailed t-test applied?

A: (i) Information describing IHC staining and the ELISA assay has been added to the Materials and Methods section. (lines 204-211; lines 125-127)

(ii) Western blot results have been added.

(iii) The ‘n’ numbers for OA patients and healthy controls have been added. (line 125)

(iv) Ethics Committee approval details and the approval number have been added. (lines 120-122)

(v) The rat was injected with control-shRNA or resistin-shRNA. Therefore, no placebo injections were administered to the control rats.

(vi) We have added details about the micro-CT ROIs into the Materials and Methods section. (lines 198-202)

(vii) The statistics section is now enriched with details about the testing, with a two-tailed test (ANOVA). Normal distribution of the data was not verified. (lines 227-232)

Q2. I know that the Authors can better (10.3390/cells9040927).

A: The manuscript has been extensively rewritten.

Minor comments (not all of them)

Q3. L72 correct to "anterior cruciate ligament transectiot"

A: “Anterior cruciate transection” has been corrected to “anterior cruciate ligament transection” (line 43)

Q4. L79 Santa Cruz is a city in CA. Santa Cruz Biotechnology headquarters are located in Dallas, TX.

A: This information has been corrected accordingly. (line 105)

Q5. L88 DMEM. While it is well-known basal medium all other abbreviations are explained.

A: DMEM has been defined accordingly. (lines 133-134)

Q6. L94 synovial tissue, synovium - please unify the nomenclature in whole manuscript.

A: “Synovial tissue” has been corrected to “synovium” for unifying the nomenclature throughout the manuscript.

Q7. L156 remove extra “mean”

A: The extra “mean” has been deleted accordingly.

Q8. L168 and others: give the exact values of concentration of resistin (1, 3, 10)

A: Detailed concentration values have been added accordingly. (line 249 and 251)

Q9. Fig. 1A – change the color of bar of the OA group.

A: The color has been changed for the OA group.

Q10. Fig 3F, 4F - please mark on the graphs that these values (0-120) show time of stimulation.

A: “(min)” has been added in Fig. 3F and 4F.

Q11. Fig 6D unit for BMD is missing.

A: The unit for BMD has been added. (Fig. 6D)

Q12. L306 Koskien et al ? ref [7] ?

A: To avoid confusion, the paragraph was re-written as follows “Koskinen et al. have documented high levels of resistin in OA synovial fluid that correlated positively with proinflammatory interleukin (IL)-6 and with the catabolic factors matrix metalloproteinases MMP-1 and MMP-3 in synovial fluid, indicating that resistin is detrimental in OA pathogenesis [7]. Similarly, we found significantly higher serum resistin levels among OA patients compared with healthy controls, which were confirmed by IHC data exhibiting higher levels of resistin expression in OA synovium than in healthy synovial tissue. Our clinical cohort yielded only serum OA samples, so we were unable to measure resistin levels in synovial fluid. Interestingly, a previous paper has described finding elevated levels of serum resistin in patients with primary knee OA compared with healthy controls [35]. We would be interested in confirming this result in future clinical investigations using OA synovial fluid samples.” (lines 414-424)

Q13. L351 references are not formatted according to journal requirements

A: The references have now been formatted according to the journal’s requirements.

Q14. L359 complete the reference data (article no 1505).

A: The reference has been corrected accordingly.

We sincerely hope that our revised manuscript is now suitable for publication in the Cells journal.

Kind regards,

Chih-Hsin Tang, PhD.

Round 2

Reviewer 1 Report

The authors answered all questions and improved the quality of the manuscript by providing information that was missing before as well as higher quality pictures and statistics.

Author Response

Dear Editor

We greatly appreciate the comments of your Reviewers on our manuscript Resistin enhances VCAM-1 expression and monocyte adhesion in human osteoarthritis synovial fibroblasts by inhibiting miR-381 expression through the PKC, p38 and JNK signaling pathways (Ref. No. cells-804877). We have carefully revised the manuscript according to the suggestions raised by the Reviewers, and specific points have been addressed as below. We have used blue font to mark up the changes in our Word file.

Reviewer 1

Q1. The authors answered all questions and improved the quality of the manuscript by providing information that was missing before as well as higher quality pictures and statistics

A: We thanks for reviewer’s comments.

We sincerely hope that our revised manuscript is now suitable for publication in the Cells journal.

Kind regards,

Chih-Hsin Tang, PhD.

Reviewer 2 Report

This paper is now much improved. In the revised version of the manuscript, the authors made significant revisions and answered all questions I rose.

However, the Authors forgot to correct the text  the numbering of the panels of Fig 5, which have changed the revised version of the manuscript (correct 5D to 5E and so on, L307-327).

After these minor corrections, the revised manuscript will be suitable for further processing.

Author Response

Dear Editor

We greatly appreciate the comments of your Reviewers on our manuscript Resistin enhances VCAM-1 expression and monocyte adhesion in human osteoarthritis synovial fibroblasts by inhibiting miR-381 expression through the PKC, p38 and JNK signaling pathways (Ref. No. cells-804877). We have carefully revised the manuscript according to the suggestions raised by the Reviewers, and specific points have been addressed as below. We have used blue font to mark up the changes in our Word file.

Reviewer 2

Q1. The Authors forgot to correct the text the numbering of the panels of Fig 5, which have changed the revised version of the manuscript (correct 5D to 5E and so on, L307-327).

A: The numbering of the panels of Fig 5 in text has been corrected. (Lines 314-325)

We sincerely hope that our revised manuscript is now suitable for publication in the Cells journal.

Kind regards,

Chih-Hsin Tang, PhD.